# Structural and Metabolic Changes in Bone

**DOI:** 10.3390/ani12151946

**Published:** 2022-07-31

**Authors:** Agata Wawrzyniak, Krzysztof Balawender

**Affiliations:** Institute of Medical Sciences, College of Medical Sciences, University of Rzeszów, Rejtana Street 16c, 35-959 Rzeszów, Poland

**Keywords:** bone, bone morphogenetic proteins, glucocorticoids, bone metabolism disorders, osteoporosis, obesity

## Abstract

**Simple Summary:**

Bone is an extremely metabolically active tissue that is regenerated and repaired over its lifetime by bone remodeling. Most bone diseases are caused by abnormal restructure processes that undermine bone structure and mechanical strength and trigger clinical symptoms, such as pain, deformity, fracture, and abnormalities of calcium and phosphate homoeostasis. The article examines the main aspects of bone development, anatomy, structure, and the mechanisms of cell and molecular regulation of bone remodeling.

**Abstract:**

As an essential component of the skeleton, bone tissue provides solid support for the body and protects vital organs. Bone tissue is a reservoir of calcium, phosphate, and other ions that can be released or stored in a controlled manner to provide constant concentration in body fluids. Normally, bone development or osteogenesis occurs through two ossification processes (intra-articular and intra-chondral), but the first produces woven bone, which is quickly replaced by stronger lamellar bone. Contrary to commonly held misconceptions, bone is a relatively dynamic organ that undergoes significant turnover compared to other organs in the body. Bone metabolism is a dynamic process that involves simultaneous bone formation and resorption, controlled by numerous factors. Bone metabolism comprises the key actions. Skeletal mass, structure, and quality are accrued and maintained throughout life, and the anabolic and catabolic actions are mostly balanced due to the tight regulation of the activity of osteoblasts and osteoclasts. This activity is also provided by circulating hormones and cytokines. Bone tissue remodeling processes are regulated by various biologically active substances secreted by bone tissue cells, namely RANK, RANKL, MMP-1, MMP-9, or type 1 collagen. Bone-derived factors (BDF) influence bone function and metabolism, and pathophysiological conditions lead to bone dysfunction. This work aims to analyze and evaluate the current literature on various local and systemic factors or immune system interactions that can affect bone metabolism and its impairments.

## 1. Introduction

Each type of tissue in a living organism has its own unique properties and functions. Bone tissue is characterized by a particular durability and mechanical strength. In addition, it has a remarkable capacity for adaptation and regeneration [1,2]. The bone that makes up the skeleton (approximately 15 percent of the body weight) is one of the largest organs of humans and animals. Between its basic functions, it also contributes to the homeostasis of our body. Bone-derived factors (BDFs) are pivotal factors that regulate the function of the skeletal system [3,4].

Histologically, the bones are classified into compact and spongy. In long bones, epiphyses are made up of cancellous bone covered by a thin layer of compact cortical bone. Compact bone forms the shafts of long bones and constitutes the outer layer of the epiphysis and all flat bones. At the microscopic level, both compact and cancellous bone typically show two types of organization: mature lamellar bone and woven bone (Figure 1A–C,E,F). Bone tissue, as a type of connective tissue, consists of bone cells and an extracellular matrix. The extracellular matrix is made up of a ground substance dominated by an inorganic part, which makes it hard and fragile, and an organic part, which gives it a slight plasticity. Bone strength is provided by collagen fibers. Bone cells are represented by osteoblasts, osteoclasts, and osteocytes [5,6].

The skeleton determines the appearance and internal structure of the organism, enables movement and respiration, protects internal organs, and provides an optimal environment for the hematopoiesis processes. Bone tissue provides solid support for the body, protects vital organs such as those in the cranial and thoracic cavities, and surrounds internal cavities containing bone marrow where blood cells are formed. Additionally, bones provide a place of attachment for the muscles, ligaments, and fascia that surround organs and protect internal organs [7]. Bone is a metabolic reserve of mineral salts (mainly calcium and phosphorus ions) and regulates their concentration in the blood, maintaining ionic homeostasis in the body. Ca^2+^ ions, crucial for all cells, are deposited in the bones if nutritional levels are satisfactory. Maintaining normal blood calcium levels involves the activity of all bone cells and is largely regulated by paracrine interactions between these and other cell types. Until now, the role of bone tissue has been attributed to mineral storage and to that of being an effector of hormones that regulate body growth (GH/IGF-1, gonadal hormones) and calcium-phosphate metabolism (PTH, calcitonin) [8]. Such a specifically constructed ground substance, mainly of inorganic compounds, leads to the maintenance of a specific metabolism and contributes to mechanical functions. Bone secretes substances in the circulation that meet the classic definition of a hormone, namely FGF23 and osteocalcin, which indicates that it is linked to the endocrine system [3,9,10,11].

Bone is a living tissue that undergoes opposing processes of remodeling and formation and destruction (resorption). The remodeling process continues throughout life, with varying intensity [12]. In adult animals, the skeletal destruction processes begin to predominate, resulting in a gradual loss of bone mineral density (BMD).

The development and maintenance of healthy bone and the regeneration of injured tissue in the human body comprise a set of intricate and finely coordinated processes [13]. Bone metabolism is a very dynamic process. This is the process of bone formation and bone reabsorption. There are many factors that can influence and control these changes [14]. Liver diseases are a frequent consequence of metabolic bone disorders and most often appear in the form of osteopenia or osteoporosis, and the first related symptom may be a pathological fracture of the weakened bone as a result of a minor trauma. The research results published so far indicate disorders of vitamin D_3_ metabolism, hyperbilirubinemia, endocrine disorders after liver damage, and the etiological factor [15,16].

## 2. The Work of Bone Tissue

Bone tissue is a specific type of connective tissue and is a complex material composed of mineral particles combined with proteins. Primarily, this specific structure provides durability. In addition, springiness must be provided so that the skeleton can mitigate shock without fracturing. If bone were made up solely of minerals, it would be more fragile and fracture more easily. If proteins dominated the structure, it would be too soft and plastic. The inorganic substances of bone tissue consist mainly of calcium phosphate (80%). Calcium carbonate, magnesium ions, sodium ions, citrates, and trace amounts of potassium, chlorine, and fluorine ions are also present. Mineral salts are deposited in bone in the form of very small crystals of dihydroxyapatite (Ca_10_ (PO_4_)6 (OH)_2_). These inorganic substances are bound in an orderly way to a matrix and give bones their hardness, rigidity, and fragility. The other very important element of the matrix is protein collagen. Other proteins of the bone can contribute to the reinforcement of the matrix. Type I collagen is synthesized in osteoblasts and is secreted and precipitated. The fibrils fuse with each other, producing larger fibers that are part of lamellar bone. Even small changes in bone shape can act on cells within the bone. Abnormalities of the collagen scaffold may be a cause for abnormal osteogenesis. Factors that cause weakening of the skeletal system can lead to disruption of mineral deposition [5,17,18].

Properly built bone provides the body with a strong skeleton that is both lightweight and strong because the bones are hollow inside. Externally, the bone is covered by a cortical part or portion, while on the inside there is a subtle mesh of interconnected lamellae that cross over each other and form the trabecular bone. Most bones are hollow structures, surrounded by a cortical sheath that is essential for their strength and protection, and a specific trabecular network within provides integrity and helps maintain strength [5,19]. The bone structure is designed to provide maximum strength without excessive volume; the shape and size of the bones can respond to different types of stresses that occur during physical activity. The bone structure is designed to provide maximum strength without excessive volume, but it should be remembered that bones are a portion of the musculoskeletal system. Therefore, muscle activity is very important for normal bone function, allowing muscles to move. Amongst the various possible types of bone connections, only joints provide flexible movement, ensuring elasticity and relieving the bones from injury and degeneration [5].

## 3. Bone Tissue Cells

Bone metabolism is controlled by a variety of environmental signals. The bone cellular compartment responds to these signals by modulating the balance between new bone formation and older bone resorption [20]. The three types of bone cells are primarily related to bone homeostasis. These are osteoblasts, osteocytes, and osteoclasts. These cell types are derived from two separate stem cell lineages. The first is the mesenchymal lineage, and the second is the hematopoietic lineage. It emphasizes the interaction between the immune system and bone and the unique regulation of bone homeostasis [1,11,21].

## 4. Osteoblasts

Osteoblasts are derived from precursor cells that can also be stimulated to become muscle, fat, or cartilage. Osteoblasts have round vesicular nuclei and basophilic cytoplasm, rich in rough endoplasmic reticulum. They are responsible for the production of a unique collagen-rich material called osteoid, the organic part of the bone matrix. If osteoblasts do not produce an extracellular matrix, they take the form of flattened cells and are spindle-shaped. Their nuclei are elongated and adhere closely to the bone surface. Their differentiation is controlled by growth factors and transcription factors: transforming growth factor (TGF-β gene β); the Cbfa 1 gene (the earliest indicator of osteogenesis); and bone morphogenetic protein 7 (BMP7). Encoding transcription factors induce proliferation, osteoblast differentiation, and the expression of osteocalcin and osteopontin [22,23,24,25,26].

In normal conditions, these cells can transform or become differentiated to form new bone. After the synthesis of the extracellular matrix is complete, the part of the osteoblasts that has been surrounded by it becomes osteocytes. By providing osteoblasts with the ability to form a calcium- and phosphorus-rich matrix, they guarantee adequate bone hardness and allow them to function properly. The activity of osteoblasts influences the deposition of inorganic bone components. Active young bone cells are found only on the surface of the bone matrix. They are associated with integrins, forming a unique layer of cuboid cells connected by adhesive and gap junctions [27].

On the surface of osteoblasts, there is a RANKL glycoprotein that can bind to the RANK glycoprotein on the surface of the osteoclast precursor. This is a way to directly contact these cells and stimulate osteoclast differentiation (Figure 2). Osteoblasts secrete proteins that initiate and regulate bone mineralization: osteonectin, osteocalcin, and hydrolases. They also secrete a protein, osteoprotegerin, which binds to RANKL and prevents osteoblast–osteoclast contact. This inhibits osteoclast precursor differentiation, stabilizes bone, and regulates bone modeling. In addition, it has an inhibitory effect on blood vessel calcification. Osteoblast activity is regulated by parathormone, thyroid hormones, growth hormone, vitamin D3, cytokines, growth and differentiation factors, and prostaglandins. Adrenal corticosteroids inhibit the activity of these cells [24,28,29,30,31,32].

In the processes of matrix synthesis and calcification, osteoblasts are cells that exhibit active protein synthesis and secretion. Among the noncollagen proteins secreted by osteoblasts is the vitamin K–dependent polypeptide osteocalcin, together with various glycoproteins, which binds to Ca^2+^ ions and concentrates this mineral locally. Osteoblasts also release membrane-enclosed matrix vesicles rich in alkaline phosphatase and other enzymes whose activity increases the local concentration of PO4^3-^ ions. In the microenvironment with high concentrations of both of these ions, matrix vesicles serve as foci for the formation of crystals of hydroxyapatite [Ca_10_ (PO_4_) 6 (OH)_2_]. The crystals grow rapidly due to the accretion of more minerals and ultimately produce a mass of the melting material [5,14].

## 5. Osteocytes

Osteocytes are long-lived cells and comprise most of all bone cells. Specially built with innate proteins that help them to survive in hypoxic conditions, osteocytes maintain biomineralization [33]. Osteocytes, encapsulated by mineralized bone matrix, are far from being passive and metabolically inactive bone cells. Instead, osteocytes are multifunctional and dynamic cells capable of integrating hormonal and mechanical signals and transmitting them to effector cells in bone and in distant tissues. Not only do osteocytes contribute to bone mass via controlling osteoblast and osteoclast activity, these cells also act as main players in phosphate metabolism [34]. Osteocytes have a flattened, large nucleus with dense chromatin, and the secretory vesicles, the Golgi apparatus, and the rough endoplasmic reticulum are poorly developed. Their thin processes are connected by gap junctions with neighboring osteocytes. Osteocytes and their processes occupy nonmineralized spaces called cavities (lacunae) and lamellae. The lacune and lamellae system provides a diffusion pathway for oxygen, nutrients, and metabolites—it communicates with spaces containing blood vessels (with vascular channels or with the marrow). The bone matrix connects osteocytes with the bone surface and the vasculature, and thereby, osteocytes directly, or by the release of effector proteins, influence osteoclast and osteoblast activity. Unlike osteoblasts and osteoclasts, osteocytes were previously defined by their morphology and location. For many years, these cells have been believed to be passive cells, and their functions have been misinterpreted. In fact, these cells have been found to have many crucial features in bone. It turns out that important the effector proteins released by osteocytes and modulating osteoblast and osteoclast formation are the molecules Sclerostin (SOST), an inhibitor of the signaling proteins (Wnt) signaling pathway, and RANKL, respectively. The dogma of osteocytes being passive by the criteria of metabolism was challenged. It proved that osteocytes are capable of bone destruction in a process termed osteocytic osteolysis and of depositing new bone material in the vicinity of osteocytes; thereby, they are involved in remodeling their immediate surrounding bone matrix [35,36]. Osteocytes are a major source of the molecules that regulate bone homeostasis by integrating both the mechanical cues and the hormonal signals that coordinate the differentiation and function of osteoclasts and osteoblasts. Osteocyte function is altered in both rare and common bone diseases, suggesting that osteocyte dysfunction is directly involved in the pathophysiology of several disorders affecting the skeleton [37]. The osteocytes are located within lacunae surrounded by a mineralized bone matrix, and their morphology differs depending on bone type. Osteocytes derive from the MSC lineage through osteoblast differentiation in four stages: osteoid-osteocyte, pre-osteocyte, young osteocyte, and mature osteocyte. As the cells transit from juvenile to adult forms, they elongate many long processes that become encircled by a calcium matrix. During these transformations, the morphological and ultrastructural changes of osteocytes occur; for example, the number of organelles decreases, and the ratio of nucleus to cytoplasm increases [38]. Osteocytes connect to themselves and to osteoblasts by a meshwork of small and numerous processes. This is extremely important and has a huge impact on bone damage and mechanical forces. Osteocytes manifest the expression of many different proteins. One of these agents has paracrine and endocrine effects that contribute to the regulation of bone remodeling. Osteocytes serve as detectors of stressor-induced bone microdamage and decide on remedial activity in osteoblasts and osteoclasts (Figure 1D) [39,40]. Advances in osteocyte biology initiated the development of novel therapeutics interfering with osteocyte-secreted molecules. Moreover, osteocytes are targets and key distributors of biological signals mediating the beneficial effects of several bone therapeutics used in the clinic. Osteocytic necrosis is caused due to pathologic conditions such as osteoarthritis and osteoporosis, leading to developing skeletal fragility and dysfunctional signal repair and/or microdamage. Immobilization-induced hypoxia and glucocorticoid treatment may also lead to osteocytic necrosis or apoptosis. Osteocytes regulate bone homeostasis and bone marrow fat via paracrine signaling, influence body composition and energy metabolism via endocrine signaling, and contribute to the damaging effects of diabetes mellitus and hematologic and metastatic cancers in the skeleton. The osteocytes are involved in bone remodeling and mineral homeostasis and review its recently discovered endocrine function. Moreover, osteocytes secrete sclerostin (a protein that works as a negative regulator of bone mass) and FGF-23, which is the most important osteocyte-secreted endocrine factor because it is able to regulate the phosphate metabolism [41,42]. Moreover, osteocytes can act as mechanosensory cells, transforming the mechanical strain into chemical signaling towards the effector cells (osteoblasts and osteoclasts). Therefore, the osteocyte plays an important role in bone biology, specifically in the remodeling process as it regulates both the osteoblast and the osteoclast activity. Notably, recent studies have deciphered aged osteocytes to have characteristics such as impaired mechanosensitivity, accumulated cellular senescence, dysfunctional perilacunar/canalicular remodeling, and a degenerated lacuna-canalicular network. Moreover inflammation, immune dysfunction, energy shortage, and impaired hormone responses possibly affect osteocytes in age-related bone deterioration [43]. Immobilization-induced hypoxia and glucocorticoid treatment may also lead to osteocytic necrosis or apoptosis. Osteocytes react to implant biomaterials in dynamic ways and are currently under active stem cell research for trauma care and bone remodeling purposes [44].

## 6. Osteoclasts

They are big cells with numerous nuclei. Osteoclasts are a type of macrophage that arise in the bone marrow, and their main function is to remodel bone. They are also required for matrix resorption during bone formation and remodeling. In areas of bone that undergo resorption, these cells on the bone surface lie in specific places known as resorption lacunae (or Howship lacunae). This place is an enzymatically etched depression or cavity in the matrix. Osteoclast development requires the macrophage colony-stimulating factor (M-CSF) and the receptor activator of nuclear factor-κB ligand (RANKL), which are produced by osteoblasts [45,46]. One of the factors that contribute to rheumatoid arthritis is tumor necrosis factor (TNF)-α. In an active osteoclast, the area of the membrane that contacts the bone forms a sealing zone called the ruffled border. This zone contains molecules that allow the transport of hydrogen ions from cells to the surface of the bone and strongly bind the cell to the bone matrix [31,47,48].

The osteoclast forms a strong and specific relationship with the bone surface through the αvβ3 integrin. It allows the transmission of signals from the matrix-derived cytoskeleton organization. This characteristic cytoskeleton generates a microenvironment between the cell plasma membrane and the bone surface. Osteoclasts secrete hydrolases that break down organic bone components and phagocytosethem. Osteoblast-secreted collagenase is absent from osteoclast hydrolases. Osteoclasts also do not have receptors for parathormone or vitamin D, and bone destruction is mediated by the RANK receptors of RANKL [31,49].

The bone homeostasis is maintained by the coupling between bone formation and bone resorbing (bone turnover). Osteocytes in adults, which make up the majority of bone cells, are differentiated osteolinear cells and are recognized as important cells that play an essential role in skeletal growth, particularly in homeostasis, bone modelling, and remodeling. Remodeling is carried out by basic multicellular units (BMU). These are tightly and spatially organized structures consisting of cooperating osteoclasts and osteoblasts. The osteoclasts located at the front of the BMU dig a resorption sinus into the bone, which is then filled with newly formed bone by the osteoblasts located further down the BMU. In spongy bone, remodeling takes place at the bone surface, and the BMUs are located under the umbrella, made up of lining cells, which maintains a microenvironment suitable for remodeling. The resorbed bone matrix releases growth factors, cytokines, and other substances that locally modify the activity of the BMU cells. Resorption and bone formation processes in the BMU are interrelated. The right balance between these processes determines the maintenance of a constant bone mass [39,50].

## 7. How Do Modeling and Remodeling Occur?

Osteoblasts and osteoclasts are specialized cells that participate in both patterning and remodeling in response to internal and external signals that can be acted upon to form or fracture bones. One way to change bone structure is through remodeling. This important local action is made possible by the interaction between osteoblasts and osteoclasts [51].

During exposure to bone damage, fluid translocates around the osteocytes, suggesting to the bone cells on the surface that their activity is to change to resorb or form bone. Inappropriate bone matrix formation associated with osteoblast failure is the cause of osteogenesis imperfecta and osteoporosis, especially that which is caused by an excess of glucocorticoids. An important cause of bone fragility is excessive bone breakdown by osteoclasts. The entire bone remodeling process of bone replacement and removal follows a strictly ordered manner and is made possible by close communication between bone cells [52,53]. All these processes are controlled by local and systemic factors, and the activation of this process is involved with the cells of the osteoblastic lineage and the precursors that will become osteoclasts. Changing the rate of production of the new osteoclasts, modifying their lifespan, or inhibiting their activity can affect the amount of bone removed. The activation and resorption phases are followed by the reversal phase. The resorbed surface is activated for the molding phase by producing a thin cement line that helps to create a strong bond between the existing bone and the newly formed bone [54]. Active remodeling can debilitate bone in many places. Osteoblasts that arrange themselves in successive layers form a very ordered structure to give the bone more strength. The final process of bone formation is the incorporation of inorganic parts into the collagen matrix. If this process is disturbed at any stage, the bone metabolism is altered and diseases occur [55,56].

The main tasks of bone remodeling are structural and metabolic changes in the functioning of the skeleton. Bone remodeling also aims to repair local damage. All changes are stimulated by the action of hormones that regulate mineral metabolism, but also by local factors. Sometimes even small loads on the skeleton can lead to micro-damage. Therefore, replacing damaged bone by remodeling restores its strength and stability. Due to the network of connections between osteocytes and osteoblasts, rapid changes occur during the processes involved [57,58].

## 8. Bone Morphogenetic Proteins

Bone morphogenetic proteins (BMPs) comprise a family of 15 cytokines involved in the growth and differentiation of various tissues and organs, such as bone, heart, kidney, eyes, skin, and teeth. BMPs are now thought to be a group of key morphogenetic messages that coordinate tissue structure across the body. The most crucial function of BMP messaging in physiology is underscored throughout the numerous roles of the dysregulation of BMP messaging in pathological mechanisms [23,59,60].

In recent years, BMP’s role in embryonic development and cell function in adult and postnatal animals has been widely studied. BMPs have been shown to play an important role in bone repair processes, contributing to their growth after bone fractures. The members of this family that influence bone remodeling stimulate the differentiation of bone marrow stem cells into bone-forming cells. BMPs are currently being tested in clinical trials for their potential to promote fracture union and bone defect healing. Among the BMPs tested is BMP 7, which also stimulates erythropoietin (EPO) production. EPO is produced in the kidney and stimulates the generation of erythrocytes from precursor cells. In the clinic, it is used for the treatment of anemia caused by chronic renal failure. Single doses of BMP have been shown to have a beneficial effect on bone healing, collagen, and carboxymethylcellulose repair, particularly in animals with preexisting estrogen deficiency and coexisting osteoporosis [61,62].

BMPs are multifunctional cytokines, participating in bone remodeling but also actively contributing to the maintenance of osteoclast homeostasis. BMPs support the various stages of osteoclast differentiation and activation through the RANK/RANKL/osteoprotegerin system, and thus, BMP signaling mediates the osteoblast–osteoclast coupling [60].

## 9. Calcium-Regulating Hormones

Bone growth and its role as a mineral store depend on the proper working of circulating hormones that interact with site-specific regulatory agents. Bone function is dependent on the action of the endocrine system, but it can also affect other organs of the body [8,63]. The main calcium-regulating hormones affect the bone supply of calcium and phosphorus and their formation and breakdown. Hormones that respond to changes in the concentration of calcium and phosphorus in the blood have an effect not only on bones but also on other organs. There is only a portion of the calcium in the food that is assimilated; some is excreted into the intestines; so, there is only a small amount of nutrition-derived calcium in the body. Although bones are permanently remodeled, the processes of breakdown and formation are the same. When the amount of calcium or phosphorus in the body is too low, regulating hormones take it from the bones to provide the proper functioning of other organs. Hormones play a very important role. The first hormones determine how much bone is created during the various stages of skeletal growth and how bone force and mass are maintained throughout life. The effects of hormones and mechanical stresses on bone are closely associated [64]. Many factors, such as lifestyle, hormones, and genes, have a significant impact on determining the maximum bone mass. The stronger the bones are during growth, the better able they are to cope with the calcium and phosphorus withdrawals that are needed and with any other changes in bone that occur with aging.

There are three calcium-regulating hormones that serve an essential role in generating healthy bones: parathyroid hormone (PTH), calcitriol, and calcitonin. The PTH, produced by the parathyroid gland, supports calcium levels, and promotes both bone resorption and bone formation. Parathyroid glands are sensitive to small changes in calcium levels and therefore control blood calcium concentrations very precisely. When calcium stores even slightly decrease, PTH secretion increases, which effects the kidneys and stimulates the production of calcitriol, which promotes intestinal calcium absorption. The parathyroid hormone-related protein (PTHrP) regulates cartilage and bone development in the fetus [65,66,67].

The second is calcitriol; the hormone is formed from vitamin D by enzymes in the liver and kidney, which stimulates the intestines to absorb enough calcium and phosphorus and also directly affects bone [68]. The key action of calcitriol aims to enhance the intestinal absorption of calcium and phosphorus, thereby providing minerals to the bones. Vitamin D deficiency leads to a disease of defective mineralization, called rickets in juveniles and osteomalacia in adults. These conditions can cause bone pain, bowing, deformities of the legs, and fractures. Vitamin D treatment can restore calcium supplies and reduce bone loss [49].

Calcitonin is a third calcium-regulating hormone produced by the parafollicular cells of the thyroid gland. Calcitonin inhibits bone disintegration and can prevent high levels of calcium in the blood. In adults, this effect may be comparatively temporary. At the beginning of life, calcitonin may be more important for bone development and the maintenance of normal blood calcium levels [10,69].

In addition to calcium-regulating hormones, sex hormones also play an extremely important role in the regulation of bone growth and the maintenance of bone mass and strength. Both estrogen and testosterone have effects on bone in both sexes [62,70]. Estrogen acts on both osteoclasts and osteoblasts to inhibit bone breakdown at all stages of life. Estrogen may also stimulate bone formation. A marked decrease in estrogen in menopause is associated with rapid bone loss. Another sex hormone, testosterone, is important for skeletal growth, due to its direct effects on bone and its ability to stimulate muscle growth; this sex hormone is also a source of estrogen in the body. It is transformed into estrogen in fat cells, and estrogen is essential for bones [71].

An influential growth regulator of the skeleton is also growth hormone (GH). It is produced by the pituitary gland and acts by stimulating the production of insulin-like growth factor-1 (IGF-1), produced in the liver. GH is essential for growth because it accelerates growth during adolescence, while with age the reduced production of GH and IGF-1 may be responsible for the inability of older people to form bone quickly [30,72].

Other very important hormones are thyroid hormones, which increase the production of energy by all cells in the body and increase the rate of bone formation as well as bone resorption. Their deficiency can affect growth in children, while an excessive amount can lead to too much bone destruction and a weakening of the skeleton [67,69].

Another hormone, cortisol (see below), the main hormone of the adrenal gland, has complex effects on the skeleton [52]. Even small amounts are necessary for normal bone development [73]. In addition to these hormones, other hormones exist and affect the skeletal system, such as insulin and leptin [74,75,76,77,78]. Leptin originates in adipocytes, including those in bone marrow. It has direct anabolic effects on osteoblasts and chondrocytes, but it also influences bone indirectly, via the hypothalamus and sympathetic nervous system, via changes in body weight and via effects on the production of other hormones (e.g., pituitary). Leptin’s role in bone physiology is determined by the balance of these conflicting effects. Systemic leptin administration in animals and humans usually exerts a positive effect on bone mass, and leptin administration into the cerebral ventricles usually normalizes the bone phenotype in leptin-deficient mice. The balance of the central and peripheral effects of leptin on bone remains an area of substantial controversy. In humans, leptin is likely to contribute to the positive relationship observed between adiposity and bone density, which allows the skeleton to respond appropriately to changes in soft tissue mass. Because both bone mineral density (BMD) and leptin concentrations are related to fat mass, it would be expected that leptin and BMD would be positively correlated, which is the finding in most studies. In the CNS-mediated effects of leptin on bone, it has become firmly established that leptin also acts directly on bone cells. The regulation of bone remodeling by an adipocyte-derived hormone implies that bone may exert a feedback control of energy homeostasis. Moreover, bone homeostasis displays a circadian rhythm with increased resorption during the nighttime as compared to the daytime, a difference that seems—at least partly—to be caused by food intake during the day. Ingestion of a meal results in a decrease in bone resorption. Gut hormones, released in response to a meal, contribute to this link between the gut and bone metabolism. The responsible hormones appear to include glucose-dependent insulinotropic polypeptide (GIP) and glucagon-like peptide-1 (GLP-1), known as incretin hormones due to their role in regulating glucose homeostasis by enhancing insulin release in response to food intake. They interact with their cognate receptors (GIPR and GLP-1R), which are both members of the class B G protein-coupled receptors (GPCRs) and are already recognized as targets for the treatment of metabolic diseases, such as type 2 diabetes mellitus (T2DM) and obesity. Glucagon-like peptide-2 (GLP-2), secreted concomitantly with GLP-1, acting via another class B receptor (GLP-2R), is also part of this gut–bone axis [79,80].

## 10. Implications of Glucocorticosteroids

Glucocorticoids (GCs) are the main hormones released by the endocrine axis; they play a very important role in both the overall growth and the maintenance of bone mass and have a significant impact on bone metabolism. GCs increase bone resorption by stimulating osteoclastogenesis by increasing the expression of the RANK ligand and decreasing the expression of its decoy receptor, osteoprotegerin. Acting through the receptor for nuclear hormones, GR, they affect the development, differentiation, and death of bone cells. GCs also affect bone regeneration and fracture healing due to their significant effect on the immune system. The most noticeable effect of GCs on bone is inhibition of bone formation. Steroid therapy induces bone mass loss and affects the transcription of certain regulatory factors that determine the bone turnover ratio. Because GCs can increase bone resorption and decrease bone formation, they consequently lead to decreased bone mass and BMD. Furthermore, GCs increase the risk of fractures of the rib and limb bones by modifying and reducing bone quality. GCs are used as anti-inflammatory and immunosuppressive drugs in severe cases of rheumatoid arthritis and other systemic diseases [52,81].

GCs are known to increase the risk of fractures and thus inhibit bone growth; so, the exposure of osteoclasts and osteoblasts to these hormones can alter the balance of activity of these cells, significantly affecting bone metabolism. Evidence suggests that the main adverse effects of elevated levels of glucocorticoids in bone are through direct actions on the cells involved in bone remodeling, that is, osteoblasts, osteocytes, and osteoclasts. There are two main effects of GC on bone metabolism: (1) the inducing of apoptosis in osteoblasts and osteocytes, thus decreasing bone formation and (2) the prolonging of osteoclast lifespan and the increasing of bone resorption.

Wnts/sclerostin are important mediators of impaired cell proliferation, increased apoptosis, altered autophagy, and changes in RANKL/osteoprotegerin (OPG) expression. The mechanism of the adverse effects of GC on bone tissue is varied and leads to reduced bone mass, altered bone structure, and increased risk of fractures and leads to complications of post-steroid osteoporosis [52]. Bone loss involves mainly trabecular bone and, to a lesser extent, compact bone, and bone mass loss is greatest during the first year of GC administration. GCs inhibit the bone formation processes, which gives bone destruction processes the upper hand and leads to a loss of bone mass. GCs have been shown to inhibit both the proliferation and the maturation of osteoblast precursors (osteoblastogenesis) and to affect the metabolic activity and survival time of osteoblasts and osteocytes, leading to apoptosis [82,83]. The presence of glucocorticoid receptors in osteoblasts, as well as in chondrocytes in the growth plate area, has also been demonstrated [84]. Through these receptors, the steroid-mediated inhibition of the replication and differentiation of these cells occurs, resulting in the synthesis of collagen and non-collagenous proteins, including osteocalcin. Children after glucocorticosteroid therapy are characterized by low growth. Regardless of the GC dose, due to the high sensitivity of the glucocorticoid receptor, there is a decrease in insulin-like growth factor (IGF-I) in the growth plate with a corresponding decrease in chondrocyte proliferation. With long-term use of GCs, receptor sensitivity decreases, as evidenced by an increase in IGF I levels. GCs also cause a decrease in IGF I synthesis in bone, which also results in their adverse effects on bone density. Insulin-like growth factor binding protein (IGFBP) is also produced in bone tissue. The most important factor that stimulates the synthesis of IGF in bone is the parahormone and, of the locally occurring factors, prostaglandin E2. IGF I increases the collagen and matrix synthesis in bone, as well as osteoblast activity and number. IGF I also acts on bone by reducing collagen degradation. GCs, by reducing IGF I production in bone tissue, cause bone degradation [52,85].

GCs also affect calcium-phosphate metabolism, reducing calcium and phosphate absorption in the small intestine by inhibiting the synthesis of calcium-binding protein (Ca BP) [15,25,39]. This action is independent of vitamin D. The calcium ion is a universally employed cytosolic messenger in eukaryotic cells and is involved in many cellular processes, such as signal transduction, contraction, secretion, and cell proliferation. The effects of Ca(^2+^) are mediated by calcium-binding proteins. They are now classified in different subfamilies as they differ in the number of Ca^2+^ binding EF-hand. To this protein belongs calbindins; there are three different calcium-binding proteins: calbindin, calretinin, and S100G. They were originally described as vitamin D-dependent calcium-binding proteins in the intestine and kidney in chicks and mammals. Under the influence of GCs, renal calcium reabsorption is also affected, leading to hypercalciuria and secondary hyperparathyroidism. At the same time, the level of calcium, inorganic phosphates, and parathormone in the blood serum remains unchanged [49,86]. GCs cause suppression of gonadal function, thus inhibiting the anabolic effect of sex hormones on bone tissue [62].

GCs regulate the metabolism of glucose, which is produced from non-sugar components, especially muscle proteins, raising blood glucose levels and thus reducing tissue sensitivity to insulin, leading to diabetes [87].

The GC-induced increase in osteoclastogenesis results in loss of bone mass. Osteoclastogenesis is stimulated by suppression of the osteoprotegerin gene, which in turn stimulates the activity of mature osteoclasts. The molecular mechanisms involved in the differentiation, maturation, and activation of antagonist cells guarantee bone integrity [88,89].

## 11. Bone Metabolism Disorders

In some pathophysiological situations, the alteration of the bone itself and the subsequent impairment of the function of the extraskeletal systems caused by abnormal bone are termed bone dysfunction. In disease, bone tissue shows a rapid loss of mass due to excessive resorption, which consequently affects osteoblast function. Some ways to inhibit osteoclast activity or promote osteoblast function by treatment with, e.g., bisphosphonates, have a beneficial effect on the outcome of the disease, indicating that enhancing bone function may be a strategy to improve the prognosis of the disease [90,91].

A heterogeneous group of drugs, including, e.g., vitamin D analogues, bisphosphonates, calcitonin, mitramycin, estrogens, synthetic parathyroid hormone fragments, and anabolic steroids, are agents that significantly affect bone metabolism. They are used in the treatment, but also in the prevention, of osteoporosis and Paget’s disease and in the treatment of hypercalcaemiain, the course of malignant tumors [52,65,71,92,93].

During intrachondral ossification, mesenchymal cells transform into chondrocytes. These cells, which will form the growth plate of cartilage, are gradually replaced by bone. Most bones are made by endochondral ossification. These include long, short, and irregular bones, but conversely, flat bones, including those of the skull, facial skeleton, and pelvis, are made by intramembraneous ossification. Through this process, mesenchymal stem cells (MSCs) transform directly into osteoblasts to organize bone. In both types of ossification, the bone formation is identical. Bone matrix synthesis begins with the construction of type 1 collagen by osteoblasts. Most proteins of the extracellular matrix of bone are type 1 collagen, which provides strength to the bone and provides a framework for the deposition of other components of the matrix, such as hydroxyapatite [94].

Bone homeostasis is controlled by various signaling pathways. The main pathways that participate in osteoblast differentiation include members of the fibroblast growth factor (FGF), the bone morphogenic protein (BMP) families, and the Wnt signaling pathway [60,95,96].

Wnt signaling is of growing interest because abnormal signaling is involved in many bone diseases and plays a key role in almost all aspects of bone development and homeostasis [97]. Understanding the functional mechanisms of Wnt signaling in bone biology and disease will help develop promising and effective therapies for the treatment of bone disease [60,96,98].

## 12. Abnormal Bone Development

Body systems that control the growth and maintenance of the skeleton can be disrupted in a variety of ways, resulting in various bone diseases and disorders. These include problems such as genetic abnormalities, developmental defects, and diseases such as osteoporosis and Paget’s disease that damage the skeleton during later life. In addition to conditions that directly affect the bones, there are many other disorders that indirectly affect the bones and thus disrupt their metabolism. Paget’s disease is a bone disease of unknown cause that involves an imbalance between the normal processes of bone formation and absorption, leading to abnormal bone formation. Some parts of the bones are overgrown and excessively thickened, while others are underdeveloped, making the bones weak and have a tendency to break (Figure 3A). Although the mechanism of the disease is not known, genetic and environmental factors are believed to contribute to the disease. The disease occurs slightly more often in men than in women, and the risk of developing the disease increases with age. In many cases, Paget’s disease is only diagnosed by chance on an X-ray. The symptoms of the disease depend on the bone affected and how advanced it is. The most common symptoms are bone pain associated with microfractures (small bone injuries that, apart from pain, usually produce no symptoms) or overload of joints and periarticular structures. In an affected bone area, the process appears to start with hyperactive osteoclasts. Knowledge of the causes and mechanisms of bone growth and development disorders allows recognition of their early symptoms. Metabolic endocrine disorders affect the entire skeleton, but early changes are best seen in the distal ends of the femurs, where growth rates are the most rapid [93,99].

Bone adapts the amount and distribution of its mass to accommodate typical mechanical loads. Thus, modeling modifies the outermost size and shape, and remodeling modifies the arrangement, size, and shape of the microscopic structures of which bone tissue is composed. The main determinants of bone histomorphology are the hardening mechanisms that allow bone to dissipate energy before disintegration. The disruption of the remodeling process due to ageing or pathological conditions can dramatically alter the microstructure of bone. The histomorphology of bone is essential to understanding how pathological conditions modify bone strength [100].

## 13. Inflammation and Cytokine Activity

Bone deterioration occurs in many patients with chronic inflammatory diseases, including rheumatoid arthritis and ankylosing spondylitis. The underlying causes of these changes are the numerous interactions between immune system function and bone metabolism. The direct impact of immune processes on bone occurs through disruption of the remodeling process and, in particular, increased bone resorption and/or impaired new bone formation. Chronic inflammation is also associated with apathy and reduced physical activity, lack of appetite, and malnutrition and leads to hypogonadotropic hypogonadism. These factors indirectly have a negative impact on the skeletal system [101].

Inflammation leads to the activation of the cells of the innate and acquired (adaptive) immune system that secrete cytokines. Some of these cytokines significantly interfere with bone tissue metabolism and are the cause of its deterioration. This is due to their influence on the differentiation and activity of osteoclasts and osteoblasts.

The main pro-inflammatory cytokines that negatively influence bone metabolism are Interleukin-6 (IL-6), Tumor Necrosis Factor-α (TNF-α), and Interleukin-1 (IL-1). Furthermore, other molecules in addition to IL-6 c have this effect, using the gp130 coreceptor, that is, Interleukin-11 (IL-11) and the leukemia inhibitory factor, as well as Interleukin-17 (IL-17), the Macrophage Colony-Stimulating Factor (M-CSF), Granulocyte-Macrophage Colony-Stimulating Factor (GM-CSF), Tumor Necrosis Factor–β (TNF-β), TGF-α and -β, epidermal growth factor (EGF), and prostaglandin E2 (PGE2) [102,103].

The effects of cytokines on remodeling can be direct and indirect. Some of them stimulate the secretion of other cytokines, acute phase proteins, proteases, and non-cytokine inflammatory mediators, leading to amplification of the signals stimulating osteoclastogenesis and inhibiting osteoblastogenesis. The final mediators of cytokine effects on osteoclasts seem to be M-CSF and the OPG/RAKL/RAK signaling pathway. The inhibitory effect on osteoblast proliferation, maturation, and activity is mediated by inhibition of RUNX2 and the WNT/β-catenin signaling pathway [104].

Macrophage colony-stimulating factor (M-CSF) is one of the key factors required for osteoclast maturation. The effect of this cytokine on bone marrow stem cells determines their initial differentiation into monocytes/macrophages, and osteoclast precursors originate from this stem cell population. The discovery of the OPG/RANKL/RANK signaling pathway was a landmark in the field of bone biology. Stimulation of the RANK receptor (TNFSF11A) causes maturation, activation, and fusion of the osteoclast precursor. Another effect of RANK stimulation is inhibition of the apoptosis of mature osteoclasts, leading to an increase in the lifespan and function of these cells. The ligand for this receptor is the RANKL protein (TNFSF11) found mainly in the cell membrane of osteoblasts, but also on the surface of immune cells, in a soluble form and embedded in the bone matrix [104].

Osteoprotegerin (OPG, TNFSF11B) is an inhibitor of the RANK-RANKL interaction. It is a so-called “decoy receptor” that, by binding a RANK ligand, reduces its availability to the receptors present in osteoclasts. This prevents the activation of osteoclastogenesis and reduces the resorption of bone tissue. In vitro, OPG inhibits the function of mature osteoclasts [105].

The role of the OPG/RNKL/RNK signaling pathway is not limited to the regulation of bone metabolism. This pathway is also involved in the development of the peripheral lymphatic system, blood vessels, and mammary glands [102].

## 14. Osteoporosis

Nutrition is important in maintaining bone health. Osteoporosis (OP), like obesity, belongs to the diseases of civilization and is one of the most frequent bone-related metabolic diseases. OP is a classic example of a multifactorial disease with a complex interplay of genetic, intrinsic, exogenous, and lifestyle factors contributing to an individual’s risk of the disease. This disease is associated with low bone mass and leads to bone tissue degradation, which, in turn, is associated with a higher risk of bone fractures. OP is a major cause of morbidity in older people [106,107].

Reduced levels of physical activity in people’s daily lives cause the development of metabolic syndromes or age-related disorders. Chronic inflammation is now understood to be an underlying pathological condition in which inflammatory cells such as neutrophils and monocyte/macrophages infiltrate into fat and other tissues and accumulate when people become obese due to overeating and/or physical inactivity. Pro-inflammatory mediators such as cytokines that are secreted in excess from inflammatory cells will not only lead to the development of arteriosclerosis when they chronically affect blood vessels but will also bring tissue degeneration and/or dysfunction to various organs. The immune system is intricately involved in bone physiology as well as pathologies. Inflammatory bone diseases such as osteoarthritis and rheumatoid arthritis are often correlated with OP and arise due to dysregulation of the homeostatic nexus between bone and the immune system, thereby leading to enhanced bone loss. Inflammatory mediators, such as reactive oxygen species (ROS) and pro-inflammatory cytokines and chemokines, directly or indirectly act on the bone cells and play a role in the pathogenesis of osteoporosis. The evidence suggests that both innate and adaptive immune cells contribute to osteoporosis. However, innate cells are the major effectors of inflammation. They sense various triggers to inflammation, such as pathogen-associated molecular patterns, damage-associated molecular patterns, cellular stress, etc., thus producing pro-inflammatory mediators that play a critical role in the pathogenesis of OP [108]. T helper (Th) cells along with various other immune cells are major players involved in bone homeostasis. Activated T lymphocytes are primary sources of the RANKL and TNF-α responsible for the bone destruction observed during various pathological and inflammatory conditions. Interestingly, T cells also possess anti-osteoclastogenic properties as the depletion of both CD4+ T and CD8+ T lymphocytes leads to decreased production of OPG [109]. Both CD4+ (Th1, Th2, Th9, Th17, Treg, NKT, and γδ T cell subsets) and CD8+ T cells play an important role in regulating bone health. Th17 cells are one of the major inducers of bone loss via the expression of higher levels of RANKL and TNF-α. On the other hand, Tregs are major inhibitors of bone loss [108,109] through the production of IL-4, IL-10, and TGF-β1 cytokines. Tregs also inhibit the effector functioning of Th17 cells in inflammation-induced bone loss. Tregs also lead to suppression of bone loss by inhibiting differentiation of monocytes into osteoclasts under both in vitro and in vivo conditions [110,111].

OP is caused by risk factors that have a significant impact on the development of the disease. Such factors include, but are not limited to, medication, underweight, hormonal disorders, calcium deficiency, inadequate vitamin supply in the diet, phosphorus deficiency, or irregular meals associated with an unbalanced diet. Proper nutrition, providing the correct amount of minerals, ensures the maintenance of adequate bone metabolism, and this is an important factor in bone health. Proper nutrition is key in both the prevention of disease and its treatment. The disease causes a decrease in bone density, resulting in a disruption of bone tissue integrity. There is an increased risk of fractures that begin to occur with activities that require exertion [112,113]. In the early stages, OP does not produce symptoms. Of all the diseases that affect the bones, OP is the most common. The structure of the ground substance of bone and characteristics such as fine-grained structure and porosity are some of the characteristics that determine bone strength. Changes in the structure or characteristic microarchitecture of bone are particularly important as they are the most common cause of fractures in OP. The places that are most at risk of developing OP are primarily the spine, wrist, and hip, as these areas are the main sites of the trabecular bone [114]. With the acceleration of population aging, the incidence of osteoporosis has gradually increased, and osteoporosis and fractures caused by osteoporosis have gradually become a serious social public health problem. The classic role of the renin-angiotensin-aldosterone system (RAAS) is to keep blood pressure stable. However, as the components of RAAS were found in bone tissues, their functions of stimulating osteoclast formation and inhibiting osteoblast activity, and thus inducing bone loss, have gradually emerged [115].

OP is distinguished by reduced BMD, disturbance of bone microarchitecture, and consequently increased bone fragility and susceptibility to fracture (Figure 3B). OP and osteoporotic fractures are becoming a serious global health problem. Calcium and vitamin D are the primary treatments for osteoporosis. However, they do not effectively reduce the incidence of fractures. Osteopenia is a condition in which BMD is lower than normal. It is characterized by a decrease in the mass of bone tissue while maintaining its ability to mineralize properly, that is, the deposition of calcium phosphates in the bones. The remodeling process continues throughout life, with varying intensity [49,116].

The most frequent form of osteoporosis is “primary osteoporosis”, which results from the cumulative effect of the loss of bone mass and the deterioration of bone structure with age. Inside the bone, there are processes of permanent, natural, forming, and destruction. The ratio of the amount of bone that is formed to that that is destroyed determines the total bone mass. These are among the factors that contribute to bone weakness, increased fracture risk, and thus to the development of OP [112,117]. Even young individuals may show signs of OP when their bones are relatively weak due to genetic defects or an inadequate diet. In young adults, breakage due to bone fragility is sporadic. Bone formation decreases with age in both sexes and usually does not follow the rate of bone resorption. The lack of balance between bone destruction and bone formation leads to loss of bone mass, contributing to the development of structural defects and creating skeletal fragility [118]. One of the first symptoms of metabolic bone disorders is bone breakdown. One of the factors contributing to both the decrease in bone loss and, thus, the prevention of fractures is the process of inhibiting bone resorption. This process can also reverse skeletal fragility. Bone mass loss can be reduced to a minimum, and OP can be predicted by proper nutrition, active exercise, and, if necessary, appropriate treatment. OP is often linked to inadequate calcium consumption, while vitamin D deficiency promotes osteoporosis by reducing calcium absorption. Proper levels of vitamin D storage maintain bone strength and may help prevent osteoporosis in older people [3]. However, calcium and vitamin D may not completely prevent bone loss and fractures. Hormone therapy can delay the onset of osteoporosis, but it has clear side effects and not all patients can tolerate it [49].

Radiography and absorptiometry are currently used to measure bone density, mass, and overall shape. They are good indicators of bone strength to assess the risk of OP. The densitometric examination is now the basic test for diagnosing osteoporosis. If OP is the result of a genetic defect, there is a characteristic pattern of marbled bone. The boundary between osteoclasts is blurred and bone resorption is dysfunctional. Bone hypertrophy and thickening occur, and myeloid cavity obstruction occurs, resulting in inhibition of blood cell formation, anaemia, and reduced leukocyte production [119]. Damaged osteoclasts in people with osteopetrosis have mutations in the genes for cellular proton pump ATP-ases or chloride channels. Dual-energy X-ray absorptiometry (DEXA) BMD testing is routinely performed on people at risk of osteoporosis [120,121].

Densitometry as an imaging method is accurate and precise (it measures bone mass with an accuracy of 1%). It is a safe examination; the dose of X-rays is 30 times lower than in chest radiography. The radiation dose is equivalent to the background radiation dose, that is, the radiation dose we receive in everyday life from cosmic radiation (the radiation spectrum includes X-rays) and from radioactive elements in the earth’s crust (radioactive elements in the earth’s crust emit ionizing radiation) [122,123].

## 15. Obesity

In recent years, obesity and osteoporosis have become important global health problems. Obesity today is one of the most serious and rapidly growing diseases of civilization. The response of the skeleton to obesity is complex and depends on various factors, such as mechanical load, type of obesity, location of adipose tissue, sex, age, bone sites, and secreted cytokines (Figure 3C) [124].

Of great importance for bone mass, metabolic disorders, and bone-related diseases is adipose tissue and its distribution. Adipose tissue is composed mainly of fat cells stored in large numbers, divided by connective tissue into fatty lobes. Its main location is in the subcutaneous tissue and around the internal organs. The main interactions between adipose tissue and bone tissue arise from the joint regulation of its function by the hypothalamus and bone marrow. The hypothalamus regulates the content of adipose tissue and bone tissue with the participation of the sympathetic nervous system. This action influences appetite regulation and modulates insulin sensitivity and, moreover, modifies the level of energy consumption and bone remodeling. In the bone marrow, adipocytes and osteoblasts originate from a common progenitor, a pluripotent mesenchymal stem cell, which has an equal propensity to differentiate into adipocytes or osteoblasts (or other lines) under the influence of several cell-derived transcription factors. This common origin of adipocytes and bone cells provides the basis for research into the mechanisms that connect these two tissues [125,126]. Adipose tissue secretes various molecules, called adipokines, which are believed to have effects on the metabolic, skeletal, and cardiovascular systems. Adipose tissue is therefore not only involved in body building and a source of energy in the body, it is also an endocrine organ and, depending on its location, plays a different role in the secretion of different amounts of specific substances. This specific type of adipose tissue is not only an energetic material, it has the ability to influence insulin sensitivity or change blood pressure levels. It has the ability to regulate endothelial function, actively participates in the inflammatory response, alters fibrinolytic activity, and can affect many pathological conditions. Adipose tissue plays a fundamental role in obesity and related diseases. Histologically, adipose tissue is divided into white adipose tissue (WAT) and brown adipose tissue (BAT) due to its different structure and function. Adipose tissue is also divided into beige adipose tissue. WAT stores energy and regulates energy metabolism and also produces and releases cytokines and hormones [127]. Over accumulation of WAT in the body can induce obesity and obesity-related diseases. The excess accumulation of WAT in the body can lead to obesity and related diseases, while BAT is involved in the generation of heat without shivering and in the generation of dietary-induced heat, being a kind of energy-intensive center [128].

Obesity is believed to affect bone health through a variety of mechanisms, including body weight, volume of adipose tissue, bone formation or resorption, pro-inflammatory cytokines, and the bone marrow microenvironment. WAT has a major endocrine function through its ability to secrete many adipokines, mainly leptin and adiponectin. Excess leptin secretion, reduced adiponectin secretion, and lower calcium absorption resulting from high fat consumption may contribute to reduced bone mass. All the data from the literature indicate that the bone response to obesity is an interconnected, multifaceted, and complex process that is dependent on a variety of factors. There are many such factors, e.g., mechanical load, type of obesity, location of adipose tissue, gender, age, bone sites and secreted cytokines. All of these factors may have an essential capacity to maintain healthy bones [129,130].

Adipocytes located in the bone marrow are a source of energy, as well as numerous substances. Adipocytes produce other biologically active substances that have integral roles in bone metabolism, including tumor necrosis factor α (TNF-α), tissue factor (TF), resistin, visfatin, and some enzymes related to steroid hormones, etc. Maintenance of a balance between bone and adipose tissue is determined by the differentiation of matrix cells into osteoblasts or adipocytes. Obesity is an important state that influences bone health by affecting changes in body weight, adipose tissue volume, bone formation or resorption, pro-inflammatory cytokines, and the bone marrow microenvironment [131,132].

Obesity studies have demonstrated the effects of WAT-secreted adipokines on bone tissue cells, as well as interactions between BAT, bone marrow adipose tissue (MAT), and bone metabolism. The relationship between adipose tissue and bones must be considered in terms of the relationship between body weight and bone density and the relationship between MAT and the age-related loss of bone mass. Some studies have shown that a high body mass index (BMI) protects against the development of osteoporosis and osteoporotic fractures in both sexes. In lean individuals with below-normal BMI, weight loss is associated with low BMD (BMD). MAT has a significant effect on bone turnover; so, determining bone marrow fat may be suitable for the diagnosis of osteoporosis [133,134,135].

Research evidence points to the multilayered and causal effects of obesity on bone health. Research studies on body weight and bone health suggest inconclusive and often contradictory conclusions. Some suggest that being overweight and obese has a beneficial effect on bone strength because it potentially reduces the risk of osteoporotic fractures and increases bone mineral density. This is known as the “obesity paradox”. On the other hand, the negative impact of excess body weight on bone quality is increasingly emphasized. The negative influence of increasing the volume of adipose tissue on BMD is especially emphasized. The reduction in bone mass in obesity may be due to the increased adipogenesis that takes place in the bone marrow. It may also be related to an increase in osteoclastogenesis, which is a result of the overproduction of pro-inflammatory cytokines.

## 16. Conclusions

Bone tissue has significant functions in the body. Disturbances in normal bone structure can significantly affect not only bone metabolism, but also the health of the entire body. Understanding the architectural, molecular, and functional biology of bone is important to better understand bone as a highly dynamic structure with diverse functions. Multifactor-dependent communication, the connection between bone tissue cells, is a key element in bone formation and resorption. Many metabolic pathways can affect bone development, homeostasis, and metabolism, ultimately leading to a number of bone diseases. Bone imaging, ultrasound, and cellular level studies, which focus primarily on morphological examination but also help in the functional assessment of bone, are helpful in the correct assessment of bone. Glucocorticoids in the skeletal system affect all types of bone tissue cells, definitively causing loss of bone mass and quality and causing a significant increase in the risk of fractures. Closely related to disorders of bone metabolism are obesity and overweight. Adipose tissue has a direct effect on bone tissue through the secretion of several cytokines. The bone marrow adipose tissue also has a significant effect on bone density and microstructure. Obesity is now regarded a disease of civilization and a multifactorial disorder. It is associated not only with excessive fat intake, but also with an inadequate balance of nutrients, e.g., the insufficient intake of vitamin D, calcium, and phosphorus. Improving bone function can have a positive impact on the whole-body system. Developing a deeper understanding of bone tissue, which is an extremely dynamic structure, will certainly help in the development of new treatments for many bone diseases.

## Figures and Tables

**Figure 1 animals-12-01946-f001:**
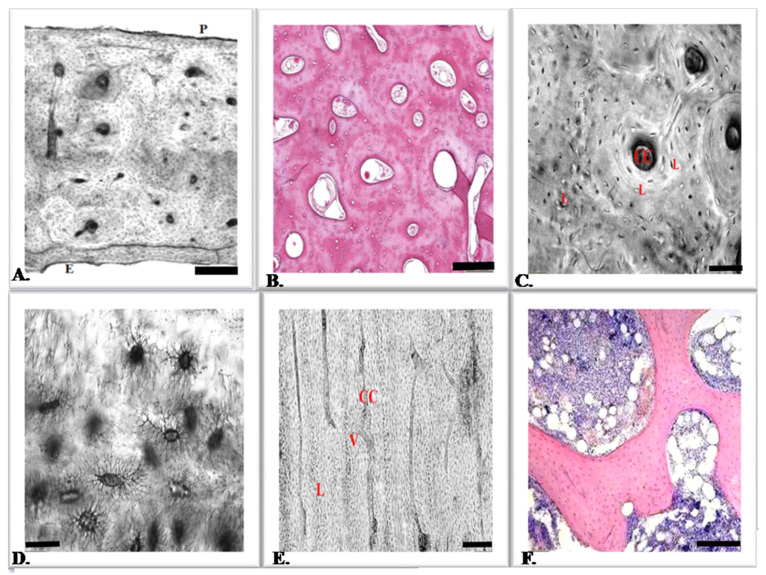
Histological structure of normal bone tissue. (**A**) A cross-section of a properly structured bone showing the compact bone. The photograph of the area of an unstained section of compact bone. Visible osteons with concentric lamellae arranged around central canals (Haversian canal). Externally visible periosteum (P), cancellous bone covered by endosteum (E). Ground bone; scale bar: 2 mm. (**B**) A cross-section of a properly structured bone showing compact bone. Because the osteons are located very close together, compact bone is stronger and harder than spongy bone. Haversian canals contain blood vessels and supply developing bone with essential nutrients. Small trabeculae create a highly porous medullary bone. They provide support, offering considerable strength without significantly increasing the weight of the bone. H & E section; scale bar: 100 µm. (**C**) Osteons are structural and functional units, characteristic of compact bone. An osteon with concentric lamellae (L) surrounding a central canal (CC) is shown. The interstitial lamellae of the osteon are also visible; these are remnants of old osteons; ground bone; scale bar: 10 µm. (**D**) Photomicrograph of osteocytes and their associated processes. Through these long processes, cell-to-cell communication takes place and substances needed by the cells are transferred. Ground bone; scale bar: 5 µm. (**E**) Longitudinal section of the compact bone. Visible Haversian canal (H) connected to the transverse perforating canal (Volkmann) (V). Osteocytes embedded with lamellae (L). Ground bone; scale bar: 200 µm. (**F**) Histological structure of spongy bone. This structure is characteristic of bone marrow. Bone trabeculae surround the bone marrow along with blood vessels, and this arrangement strengthens the spongy bone, making it stronger. H & E section; scale bar: 50 µm.

**Figure 2 animals-12-01946-f002:**
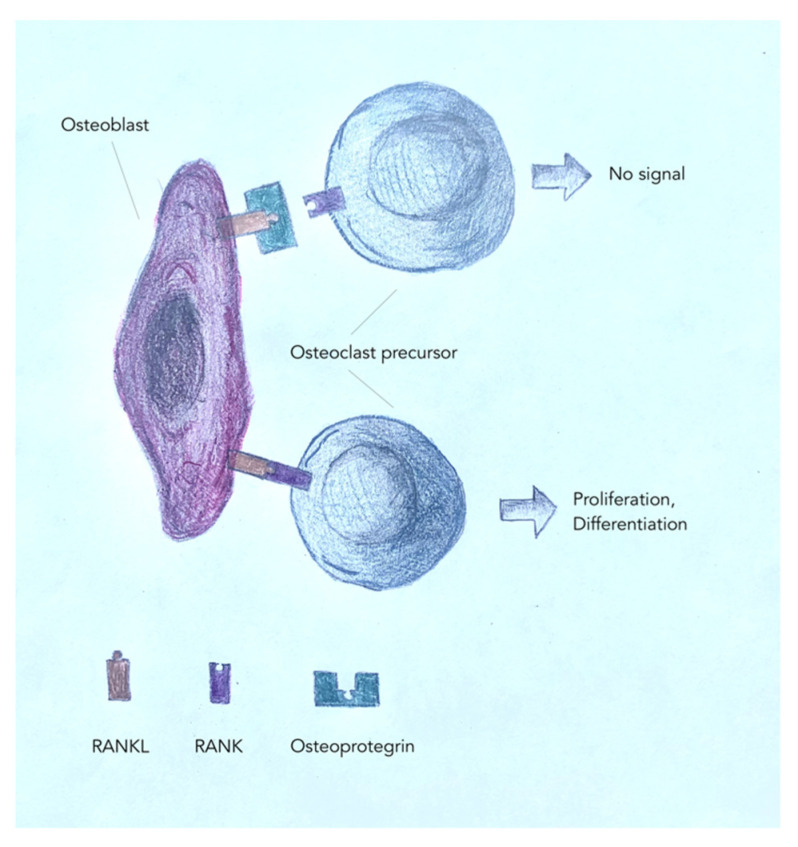
Mechanisms of osteoblast/osteoclast regulation.

**Figure 3 animals-12-01946-f003:**
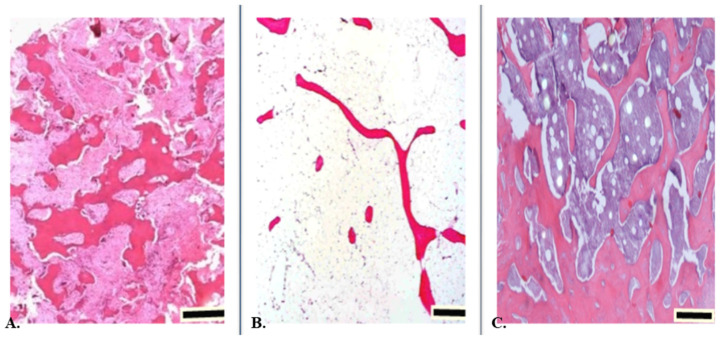
Histological structure of altered bone tissue. (**A**) Paget’s disease is a bone disorder. Histologically, the bone appears mosaic in appearance. It has thick cement lines that demarcate the disorderly lamellar bone. Such extensive trabeculae and lack of organization of the cortex are the cause of the loss of resistance to strain and therefore greatly increase the susceptibility to fracture. H & E section; scale bar: 200 µm. (**B**) In osteoporotic bone, there is an increased amount of adipose tissue appearing in the intertrabecular area. Therefore, for the same volume, the bone has much less density but is more frangible. Increased medial adiposity, associated with intratrabecular connectivity, emphasizes the functional connection between bone and marrow. Thin trabeculae are separated from each other. H & E section; scale bar: 100 µm. (**C**) Longitudinal section of the cancellous bone of the obese rat. Visible network of trabeculae separated by intercellular spaces containing islands and clusters of haematopoietic cells admixed with adipocytes. H & E section; scale bar: 500 µm.

## Data Availability

All data discussed in this article are publicly available from other sources.

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
