# Peer review of "Structural and Metabolic Changes in Bone"

_animals, 2022, doi:10.3390/ani12151946_

Round 1
Reviewer 1 Report
The authors carry out a narrative review of bone physiology in the first part of the article focusing on bone remodeling. In the second part, they analyze the influence of various pathological situations on bone remodeling.
Recommendations 1. Making a graphic representation of bone remodeling with interaction between osteoblasts and osteoclasts could be interesting2. The pathophysiology of osteoporosis should be expanded with more reference to the autocrine factors involved.
Author Response
Dear Mr/Mrs Reviewer
We would like to thank you very much for your thorough evaluation of our work, your positive feedback, as well as your critical comments. They provide important guidance to improve the quality of our future research work. We have responded to them below, with comments we fully agree with
- Thank you to the reviewer for her/his attention to the graphical representation of bone remodelling including the interaction between osteoblasts and osteoclasts. Drawing, was made and included in the text
- The pathophysiology of osteoporosis has been expanded to include greater reference to the autocrine factors involved.
We hope that the improved work will meet your expectations.
With respect,
Agata Wawrzyniak,Krzysztof Balawender
Reviewer 2 Report
The aim of the reviewed work was to analyze and evaluate the current literature on various local and systemic factors or immune system interactions that can affect bone metabolism and its impairments. In the available literature, there are no publications describing the issues of structural and metabolic changes in bones in this approach. Minor typing errors were noticed. For example: line 226 - there is: "phagocytosethem", correctly: "phagocytose them"; line 485 - there is "adiposse tissue", correctly: "adipose tissue"; line 689 - there is: "boneformation", correctly: "bone formation". In conclusion I think that the review paper may be published in the journal Animals.
Author Response
Dear Mr/Mrs Reviewer
We would like to thank you very much for your thorough evaluation of our work, your positive feedback, as well as your critical comments. They provide important guidance to improve the quality of our future research work. We have responded to them below, with comments we fully agree with.
All typing errors have been corrected
We hope that the improved work will meet your expectations.
With respect,
Agata Wawrzyniak, Krzysztof Balawender
Reviewer 3 Report
Comments to the Authors
Manuscript ID: animals-1796725
Title: Structural and metabolic changes in bone
This work aims to analyze and evaluate the current literature on various local and systemic factors or immune system interactions that can affect bone metabolism and its impairments. Understanding the architectural, molecular, and functional biology of bone is important to better understand the bone as a highly dynamic structure with diverse functions. Developing a deeper understanding of bone tissue will certainly help in the development of new treatments for many bone diseases.
The review article should contain the latest published data. In this case, there are cited 107 publications, and only 7 articles have been published in the period of 2018-2022. I recommend to complete the article with the latest data.
There is too little information about osteocytes. Only 3 publications have been cited. Unfortunately, I have the impression that this is mainly textbook knowledge.
The authors mention the influence of insulin and leptin on the skeletal system. How do insulin and leptin affect the skeletal system? (line 360)
In the sentence:
,,GCs also affect calcium-phosphate metabolism, reducing calcium and phosphate absorption in the small intestine by inhibiting the synthesis of calcium-binding protein”. Which calcium-binding protein it is referred to?
Line 413 - reference numbers should be placed in square brackets [ ].
The subject of review article is adequate to its content. The article is written in a clear and understandable way.
The only difficulty is the appearance of abbreviations in the text for the first time, which are explained later, making it difficult to read the work. For example:
line 386 - osteoprotegerin (OPG), when the abbreviation has already been used - line 297.
Please explain the abbreviations used in the text. For example: GM-CSF, TNF-β (line 516).
There is no scale bar in Figure 1 (A-E).
There is an unnecessary dot at the end of the sentence (line 438, 547, 556, 655).
Please use a space to separate words or numbers in the literature.
Line 887 (reference 79) - no year
In my opinion, the manuscript may be ready for publishing after the recommended corrections.
Author Response
Dear Mr/Mrs Reviewer
We would like to thank you very much for your thorough evaluation of our work, your positive feedback, as well as your critical comments. They provide important guidance to improve the quality of our future research work. We have responded to them below, with comments we fully agree with.
- Thank you to the reviewer for pointing out that there is too little information on osteocytes. The section on osteocytes has been expanded and updated with more recent literature
- Thank you to the reviewer for pointing out that the number of recent citations was too low. After rewording the paper, the number of recent citations has increased (the text has been supplemented by 28 current items)
- Information on the effects of insulin and leptin on the skeletal system has been explained, similarly on the calcium-binding protein.
- The abbreviations in the text have been developed
- Figure 1. has been corrected. Scal bar is now visible
- All other suggested changes have been made to the text
We hope that the improved work will meet your expectations.
With respect,
Agata Wawrzyniak, Krzysztof Balawender
Round 2
Reviewer 1 Report
The authors have made a great effort to explain the physiology of bone and its alteration in metabolic bone disease. Although the study does not provide important novelties, it can help to understand these mechanisms.
The suggestions made have been resolved